# Impacts of Intercropped Maize Ecological Shading on Tea Foliar and Functional Components, Insect Pest Diversity and Soil Microbes

**DOI:** 10.3390/plants11141883

**Published:** 2022-07-20

**Authors:** Yan Zou, Fangyuan Shen, Yanni Zhong, Changning Lv, Sabin Saurav Pokharel, Wanping Fang, Fajun Chen

**Affiliations:** 1Department of Entomology, College of Plant Protection, Nanjing Agricultural University, Nanjing 210095, China; 2020202034@stu.njau.edu.cn (Y.Z.); 2019802143@njau.edu.cn (F.S.); 2020802143@stu.njau.edu.cn (Y.Z.); timber1013@163.com (C.L.); pokharelsabin93@gmail.com (S.S.P.); 2Department of Tea Science, College of Horticulture, Nanjing Agricultural University, Nanjing 210095, China

**Keywords:** ecological cultivation mode, ecological shade, foliar nutritional, functional components, population occurrence, community diversity, insect pests, soil microbes

## Abstract

Ecological shading fueled by maize intercropping in tea plantations can improve tea quality and flavor, and efficiently control the population occurrence of main insect pests. In this study, tea plants were intercropped with maize in two planting directions from east to west (i.e., south shading (SS)) and from north to south (i.e., east shading (ES) and west shading (WS)) to form ecological shading, and the effects on tea quality, and the population occurrence and community diversity of insect pests and soil microbes were studied. When compared with the non-shading control, the tea foliar nutrition contents of free fatty acids have been significantly affected by the ecological shading. SS, ES, and WS all significantly increased the foliar content of theanine and caffeine and the catechin quality index in the leaves of tea plants, simultaneously significantly reducing the foliar content of total polyphenols and the phenol/ammonia ratio. Moreover, ES and WS both significantly reduced the population occurrences of *Empoasca onukii* and *Trialeurodes vaporariorum*. Ecological shading significantly affected the composition of soil microbial communities in tea plantations, in which WS significantly reduced the diversity of soil microorganisms.

## 1. Introduction

Tea, *Camellia sinensis* (L.), is a subtropical perennial evergreen crop, and it is one of the main cash crops in China [1] and commercially cultivated in more than 60 countries worldwide [2]. Tea plants are prominent sciophytes, and are suitable for growing under diffused light conditions. Due to the frequent high temperatures and intense light weather in hilly tea plantations, the photo-inhibition of tea plants occurs, resulting in the decline of tea quality [3]. Intercropping systems, such as rubber–tea, chestnut–tea and other tea–forest intercropping can reduce the strong light and temperature in tea plantations, and thus the produced green tea is exotic in taste and more desirable [4]. Shading can affect the photosynthesis and metabolites of tea plants, and change the tea quality, which would affect the occurrence of herbivorous insects in the tea plantation [3,5]. Teng et al. (2021) indicated that the shading of tea by tall trees was beneficial to reduce lignin synthesis and improve tea quality [3]. Qin et al. (2011) documented that the use of shading increased the foliar contents of amino acids (AAs) and caffeine, while it reduced the foliar contents of tea polyphenol (TP) and the TP/AA ratio [5]. It is generally believed that a high content of amino acids significantly improves the freshness and aroma of tea leaves, and that an appropriate phenol/ammonia ratio is a necessary condition for the production of high-quality tea [6,7].

Intercropping is one of the main measures for habitat management and pests’ control in tea plantation [4,8,9,10]. Many studies have shown that intercropping in tea plantations is beneficial to improve the leaf yield and quality of tea plants. De and Surenthran (2005) indicated that the incorporation of contour hedgerows had the potential to regenerate soil fertility and sustain tea yields on sloping terrains [11]. Farooq et al. (2021) revealed that groundnut–tea intercropping could enhance soil nutrients’ status and positively impact soil conservation, and was also beneficial for increasing biological diversity and ecological stability [12]. Diverse studies showed that the ecosystem of the artificial compound intercropping in tea plantations had higher biodiversity and a more stable insect community structure than that of the pure tea plantation [13,14]. The higher the community diversity is, the stronger the stability of the entire tea plantation ecosystem will be. The high biodiversity of the compound tea plantation can effectively improve the anti-interference capability of the whole tea plantation ecological system, and the predator abundance of insect pests in intercropping tea plantation was much higher than that in pure tea plantations, which was a favorable effect [8]. In addition, ecological shading could cause changes in soil microbial communities [15].

Here, we designed the ecological shading experiment by intercropping maize in two planting directions from east to west (i.e., forming south shading) and from north to south (i.e., forming east shading and west shading, in order to find a more economical and high-quality cultivation mode with the adoption of sustainable ecological management tactics for the production of an enhanced quality of tea. Moreover, the effects of ecological shading by intercropping maize on tea quality and flavor, as well as the population abundances and community diversity of insect pests and the soil microbes in the tea plantation, were studied vividly.

## 2. Materials and Methods

### 2.1. Experimental Site Description

The experimental tea plantation was located in Hongqi Village, Jiangning District, Nanjing City, Jiangsu Province of China (31.72° N, 118.75° E). The selected tea species was *Camellia sinensis* cv. Huangshan zhong. This tea plantation has been continuously produced for more than 10 years. The average annual temperature in this area is about 16 °C, the average annual precipitation is about 1073 mm, and the average annual frost-free period is 224 days.

### 2.2. Maize Intercropping Setup

The tea plants were planted in two directions in the tea plantation, i.e., east–west rowing direction and south–north rowing direction, with a row spacing of 1.5 m and a plant spacing of 0.5 m. In June of 2019 and 2020, two rows of maize plants (cv., Yunong Jingtiannuo from Jiangxi Yufeng Seed Industry Co., Ltd. Xinyu, China) were intercropped between two rows of tea plants, with a row spacing of 0.10 m and a plant spacing of 0.20 m to form an ecological shade in tea plantations. Two rows of corn were planted alternately, and the close distance between the two corn plants of these two rows was 0.14 m. For the tea plants in the north–south rowing direction, the intercropped maize plants formed an east shading (i.e., ES) and a west shading (i.e., WS) for the tea plants on the west and east side of the intercropped maize plants, respectively. For the tea plants in the east–west rowing direction, the intercropped maize plants formed a south shading (i.e., SS) for the tea plants on the north side of the intercropped maize plants. The tea plants in the north–south rowing direction without intercropped maize were used as the control for the ecological shading treatments of ES and WS (i.e., EWCK), and those in the east–west rowing direction without intercropped maize were used as the control for the ecological shading treatment of SS (i.e., SSCK). There were a total of five treatments of ecological shading (i.e., ES, WS and their control of EWCK, and SS and its control of SSCK), and each ecological shading treatment was repeated five times (Figure 1). A total of 25 investigation plots were randomly set in the area suitable for the experimental conditions in the tea plantation. Each plot comprised two 6 m in length rows of tea trees and two 6 m in length rows of maize corresponding to the tea trees (or without maize).

After the maize plants grew higher than tea plants, the ecological shading experiment began, and the uppermost tender leaves of tea plants were collected on 15 August, 15 September, and 15 October of 2019 and 2020. Additionally, the field layout of the ecological shading treatments by intercropping maize plants in the tea plantation is shown in Figure 1. During the whole period of the experiment, no additional farming operations were carried out in the experimental plots. A photometer (Model: 1801C; Delixi Electric LTD, Leqing, Zhejiang Province of China) was used to measure the light intensity on the tea canopy of different ecological shading treatments and the respective control treatments on three sunny days in order to show the effects of intercropped maize ecological shading on the light intensity on the canopy of the tea plants (a supplemental measurement in that early stage of shade formation; the light intensities of the shading treatment and control were measured every 30 min. The measuring point was the top leaf of the tea tree, the detection points were random, and the number was 9). According to the actual observed shading time, the measurement was carried out in different time periods. The light intensity of the tea canopy for the treatments of ES and its control, EWCK, was measured from 6:00 a.m. to 10:30 a.m. (i.e., from the sun rise to straight-lighting time), that of the treatments of WS and its control, EWCK, was measured from 2:00 a.m. to 6:00 p.m. (i.e., the sun straight-lighting time to sundown), and that of the treatments of SS and its control, SSCK, was measured from 11:00 a.m. to 1:30 p.m. The ecological shading treatment significantly reduced the light intensity on the canopy of the tea plants (ES: *F* = 1625.21, *p <* 0.001; WS: *F* = 679.97, *p <* 0.001; SS: *F* = 56.53, *p <* 0.001; Appendix A Table A1).

### 2.3. Determination of Nutritional and Functional Components in Tea Leaves

#### 2.3.1. Foliar Nutrient Contents

*Soluble sugar content* The test kit of plant soluble sugar contents (No. A145-1-1; Nanjing Jiancheng Bioengineering Institute, Nanjing, China) was used to measure the foliar soluble sugar content of the tested tea leaves. The determination principle is that after the sugar is mixed with concentrated sulfuric acid solution, the obtained product, furfural or hydroxymethyl furfural, could react with anthrone, and the color of the furfural derivative is in direct proportion to the sugar content. Therefore, the absorbance value can then be determined at 630 nm to determine the content of soluble sugar in the sample.

*Soluble protein content* The total protein quantitative assay kit (No. A045-2; Nanjing Jiancheng Bioengineering Institute, Nanjing, China) was used to measure the foliar soluble protein content of the tested tea leaves. The principle is that a protein molecule has an amino group; when Coomassie brilliant blue is added into a protein standard solution or a sample, anions on the Coomassie brilliant blue dye can be combined with the protein amino group to change the solution into blue, and the protein content can be calculated by measuring the absorbance at 595 nm.

*Free fatty acid content* The non-esterified free fatty acids assay kit (A042-1-1; Nanjing Jiancheng Bioengineering Institute, Nanjing, China) was used to measure the foliar free fatty acids’ content of the tested tea leaves. The free fatty acids can combine with copper ions to form a copper salt of fatty acid and can be dissolved in chloroform, and the content of free fatty acids can be calculated by determining the content of copper ions in the copper reagent.

#### 2.3.2. Foliar Functional Component Contents

*Polyphenols content* The phenolic content of the tea leaf extracts was determined by using the Folin–Ciocalteu colorimetric method [16]. All sample extracts were diluted 1:20 with distilled water to obtain readings within the standard curve ranges of 0.0–600.0, 1 g of gallic acid per milliliter. Tea leaf extracts were oxidized with the Folin–Ciocalteu reagent, and the reaction was neutralized with sodium carbonate. The absorbance was measured at 760 nm after 90 min at room temperature by an MRX II Dynex plate reader (Dynex Technologies, Inc., Chanilly, VA, USA) (Li et al., 2019). The absorbance values were then compared with those of standards with known gallic acid concentrations.

*Caffeine content* The foliar caffeine content of the tea plants was quantified using an HPLC-based method [17]. The leaf samples were dried at 80 °C for 24 h for the determination of caffeine content in the tea leaves. Then, the caffeine was extracted and purified from the drying tea leaves.

*Theanine content* Theanine was determined by adding 5 mL of the tea extract and 5 mL of sulfo-salicylic acid, and the mixture was centrifuged at 13,000 rpm for 5 min to promote the reaction and then filtered using a 0.20 µm nylon membrane filter followed by an amino acid analyzer (Hitachi L-8900, Tokyo, Japan) [18,19].

*Catechin content* Catechin in tea leaf samples was extracted with a 70% methanol aqueous solution in a 70 °C water bath. The determination of catechins was performed on a C18 column with a detection wavelength of 278 nm, gradient elution, HPLC analysis [20].

#### 2.3.3. Leaf Quality Indexes

Two leaf quality indexes were measured, including the catechin quality index and phenol/ammonia ratio. The relative correction factor of catechins and caffeine in the results of ISO international environmental test was used for quantification, i.e., the formula of catechin quality index = [EGCG (%) + ECG (%)]/EGC (%) × 100 (here, EGCG—Epigallocatechin gallate; EGC—Epigallocatechin; ECG—Epicatechin gallate) [21]. The phenol/ammonia ratio, i.e., the ratio of tea polyphenols to amino acids, is an important indicator for evaluating the quality of green tea, and the formula of the phenol/ammonia ratio = total polyphenol content (mg/g)/amino acid content (mg/g).

### 2.4. Insect Investigation

The field investigation was conducted every 10 days after the intercropped maize plants were taller in height than tea plants, i.e., from the last ten days of July to the last ten days of October in 2019 and 2020, respectively. Three tea plants were randomly selected for each treatment of the ecological shading (including ES, WS and SS) and its respective control (WECK for ES and WS and SSCK for SS) to count the number of insects, and the species of collected insects were also identified and classified. Cameras, insect nets, and suction-implements were used as methods of insect collection and statistics. In this experiment, two key insect pests, *Empoasca onukii* and *Trialeurodes vaporariorum*, as being the main insect pests in the tea plantation, were selected to assess the effects of ecological shading by intercropping maize on the population occurrences of insect pests. Moreover, the Shannon–Wiener index (*H*), Pielou evenness index (*E*), Margalef richness index (*D*) and Simpson dominance index (*C*) of the insect community were calculated based on the species and numbers of sampled insects on tea plants. The formulas are as follows:

Shannon–Wiener diversity index:(1)$$H=-\sum_{i=1}^{S}P_{i}\times \ln(P_{i})\;P_{i}=N_{i}/N$$

Pielou evenness index:(2)$$E=H/H_{\max}\;H_{\max}=\ln S$$

Margalef richness index: (3)D=(S−1)/lnN

Simpson dominance index:(4)$$C=\sum_{i=1}^{S}{\left(P_{i}\right)}^{2}\;P_{i}=N_{i}/N$$

*P_i_*: relative abundance of insect species *i*; *N_i_*: number of individuals for species *i*; *N*: the total number of individuals of all species in the community; *S*: the number of species in the community; *H*_max_: Maximum species diversity index.

### 2.5. Composition and Diversity of Soil Microbial Community in the Tea Plantation

On 2 October 2020, three sampling sites were randomly selected in each treatment of ES, WS, SS, WECK and SSCK, and the surface soil (depth: 0–20 cm) near the tea tree plants was collected. The collected soil samples were sent to Shanghai Personal Biotechnology Co., Ltd. for the 16S rRNA gene sequencing study. Initial screening was performed on the original off-board data of high-throughput sequencing based on sequence quality and retested the problem samples. The library and samples were divided according to the index and barcode information of the original sequence of an initial quality screening, and then the barcode sequence was removed. We performed sequence denoising or operational taxonomic units (OTUs) clustering, according to the QIIME2 DADA2 analysis process or V-search software analysis process. The specific composition of each sample (group) at different taxonomic levels was displayed to understand the overall situation. In order to compare the differences in microbial composition among the samples and display the distribution trend of species abundance of each sample, a heat map was used for species’ composition analysis. The heat map was drawn using abundance data from the top 20 genera with an average abundance. The horizontal and vertical coordinates of the heat map plotted the clustering tree ordering according to the correlation between the samples, i.e., plotted the clustering heat map. The diversity indices of soil microbial communities for different samples were counted, including the *Chao1* index, Shannon–Wiener diversity index (*H*), Pielou evenness index (*E*), and Simpson dominance index (*C*). The latter three formulas were same as in Section 2.4. In addition, for the Chao1 indices (*Chao1* = *S* + F_1_^2^/2F_2_), F_1_ and F_2_ are the count of singletons and doubletons, respectively.

### 2.6. Data Analysis

SPSS 25.0 (IBM Corporation, Armonk, NY, USA) and GraphPad Prism 7 (GraphPad Software, Inc., San Diego, CA, USA) were used for the statistical analysis. The latter was used for making line and column charts. A two-way repeated measures ANOVA was used to analyze the effects of sampling year (2019 vs. 2020), ecological shading treatment (SS, ES, WS, SSCK and EWCK) and their interaction (with sampling time as repeated measures) on the measured indexes of foliar soluble nutrients, functional components and quality indexes of the tea leaves, the population dynamics (individuals per plants) of the key species of *E. onukii* and *T. vaporariorum*, and the dynamic values of the community diversity indexes (*H*, *E*, *D* and *C*) for the insects in the tea plantations. Additionally, one-way ANOVAs were used to analyze the effects of ecological shading treatment on the diversity indices of soil microbial communities in the tea plantation. Furthermore, the significant differences between/among treatments were analyzed by the *LSD* test or *t*-test at *p* < 0.05.

## 3. Results

### 3.1. Effects of Intercropped Maize Ecological Shading on Foliar Soluble Nutrients of Tea Plants

The ecological shading treatment and sampling year both had significant effects on the foliar contents of soluble sugars and free fatty acids, while it had no significant effect on the foliar content of the soluble protein in tea plants (Table 1). When compared with the respective control, the no-shading treatment (SSCK or EWCK), south shading (SS), east shading (ES) or west shading (WS) did not significantly affect the foliar content of soluble sugar, while ES and WS both significantly decreased the foliar content of the free fatty acids of the tea plants (Figure 2).

There were no significant differences in the contents of the foliar soluble nutrients of the tea plants among the three ecological shading treatments of SS, ES and WS (Figure 2). Additionally, the foliar contents of soluble sugars and free fatty acids of tea plants in the control no-shading treatment of SSCK (i.e., the east–west rowing direction) were significantly lower than those in the control no-shading treatment of EWCK (i.e., the north–south rowing direction) (Figure 2).

### 3.2. Effects of Intercropped Maize Ecological Shading on Foliar Functional Components of Tea Plants

Ecological shading treatment, sampling year and their interaction had significant effects on the foliar contents of the functional components (including polyphenols, caffeine and theanine) of tea plants. When compared with the respective control no-shading treatment (i.e., SSCK or EWCK), the ecological shading treatments of SS, ES and WS significantly reduced the foliar content of polyphenols, and significantly enhanced the foliar contents of caffeine and theanine in the tea plants (Figure 3).

The foliar polyphenol content of the tea plants in the control no-shading treatment of EWCK was significantly higher than that in the control no-shading treatment of SSCK, and the trend was just opposite for the foliar theanine content, while there was no significant difference in the foliar caffeine content of tea plants between the two control no-shading treatments of SSCK and EWCK (Figure 3). Moreover, the foliar contents of polyphenols and caffeine of tea plants in the ecological shading treatments of ES and WS were significantly higher than those in the ecological shading treatment of SS, and the tendency was just opposite for the foliar theanine contents, which in the ecological shading treatment of ES were significantly higher than in the ecological shading treatment of WS (Figure 3).

### 3.3. Effects of Intercropped Maize Ecological Shading on the Leaf Quality Indexes of Tea Plants

Ecological shading treatment, sampling year and their interactions had significant effects on the leaf quality indexes (i.e., catechin quality index and phenol/ammonia ratio) of tea plants, except for the effect of sampling years on the phenol/ammonia ratio of tea leaves. When compared with the respective control no-shading treatment (SSCK or EWCK), the ecological shading treatments of SS, ES and WS significantly increased the catechin quality index, and significantly decreased the phenol/ammonia ratio in the leaves of the tea plants (Figure 4). Moreover, the values of the catechin quality index and phenol/ammonia ratio in the leaves of the tea plants in the control no-shading treatment of EWCK were significantly higher than those in the control no-shading treatment of SSCK (Figure 4). Furthermore, the values of these two leaf quality indexes of the tea plants in the ecological shading treatments of ES and WS were significantly higher than those in the ecological shading treatment of SS respectively, and the phenol/ammonia ratio in the leaves of tea plants in the ecological shading treatment of WS was significantly lower than that in the ecological shading treatment of ES (Figure 4).

### 3.4. Effects of Intercropped Maize Ecological Shading on Population Dynamics of Key Tea Pests and Community Diversity in Tea Plantations

#### 3.4.1. Population Dynamics of Key Insect Species

Overall, the ecological shading treatment, sampling year and their interaction had significant effects on the population dynamics of *T. vaporariorum* and *E. onukii* on tea plants, except for the effect of sampling year on the population dynamics of *E. onukii* fed on tea plants. When compared with their respective control no-shading treatment (SSCK or EWCK), the ecological shading treatments of ES and WS significantly reduced the population dynamics of *T. vaporariorum* and *E. onukii* on tea plants, while the ecological shading treatment of SS significantly increased the population dynamics of *T. vaporariorum* on tea plants (Figure 5). Additionally, the population dynamics of *T. vaporariorum* and *E. onukii* on tea plants in the ecological shading treatments of ES and WS were significantly lower than those in the ecological shading treatment of SS, respectively (Figure 5). Moreover, the population dynamics of *T. vaporariorum* and *E. onukii* on tea plants in the control no-shading treatment of SSCK were significantly lower than those in the control no-shading treatment of EWCK (Figure 5). 

#### 3.4.2. Community Diversity of Insects

Ecological shading treatments and sampling year both significantly affected the values of the Shannon–Wiener index (*H*). Additionally, the ecological shading treatment significantly influenced the values of the Margalef richness index (*D*), and the sampling year significantly impacted the values of the Simpson dominance index (*C*), describing the insect community in tea plantation (Table 1).

When compared with their respective control no-shading treatment (SSCK or EWCK), the ecological shading treatments of ES significantly enhanced the value of the Shannon–Wiener index (*H*), and the ecological shading treatments of ES and WS both significantly increased the value of the Margalef richness index (*D*) of the insect community in the tea plantation (Figure 6). Additionally, the value of the Shannon–Wiener index (*H*), describing the insect community in the ecological shading treatment of ES, was significantly higher than that in the ecological shading treatment of SS (Figure 6). There were no significant differences in the values of all four community diversity indexes of insects between the controls (no-shading treatments of SSCK and EWCK) (Figure 6).

### 3.5. Effects of Intercropped Maize Ecological Shading on the Community Structure and Diversity of Soil Microorganisms in Tea Plantation

#### 3.5.1. Taxonomic Composition of Soil Microorganisms

Figure 7 is a heat map showing the average abundance data of the top 20 genera in the soil samples from the ecological shading treatments (SS, ES and WS) and their respective control no-shading treatment (SSCK and EWCK), and it indicated a distinct soil bacterial composition between/among the ecological shading treatments and their control no-shading treatment (i.e., SS vs. SSCK; ES and WS vs. EWCK). *Burkholderia-Caballeronia–Paraburkholderia*, *KF-JG30-C25*, *Bradyrhizobium* and *Rhodanobacter* were the four dominant bacterial genera in the ecological shading treatments (SS), and *Acidothermus*, *Acidibacter*, *AD3*, *Subgroup_2*, *WPS-2* and *IMCC26256* were the six dominant bacterial genera (Figure 7); *Granulicella*, *Saccharimonadales*, *Psdudolabrys*, *Acidipila*, and *Chujaibacter* were the five dominant bacterial genera in the ecological shading treatment of WS, and *Bryobacter*, *Candidatus_Solibacter*, *Subgroup_6*, *Haliangium*, and *Ellin6067* were the five dominant bacterial genus in the control non-shading treatment of EWCK, and there were no dominant bacterial genus in the ecological shading treatment of ES (Figure 7).

The relative abundance of soil microorganisms at the phylum level was analyzed in Figure 8. The relative abundance of *Proteobacteria*, *Bacteroidetes* and *Verrucomicrobia* was significantly increased by 116.2, 490.4 and 599.7%, respectively, and that of *Acidobacteria*, *Actinobacteria*, *Gemmatimonadetes* and *WPS-2* was significantly decreased by 54.3, 50.7, 31.2 and 94.4% in the ecological shading treatment of SS when compared with the control no-shading treatment of SSCK, respectively (Figure 8). Moreover, the relative abundance of *Bacteroidetes*, *Firmicutes* and *Patescibacteria* was significantly increased by 1.6, 4.3 and 10.3 times in the ecological shading treatment of WS, and the relative abundance of *WPS-2* was significantly increased by 11.1 and 7.6 times in the ecological shading treatments of ES and WS when compared with the control no-shading treatment of EWCK, respectively (Figure 8). There were no significant differences in the relative abundances of the soil microorganisms at the phylum level in the ecological shading treatment of ES when compared with that in the control no-shading treatment of EWCK, respectively (Figure 8).

#### 3.5.2. Community Diversity of Soil Microbial Microorganisms

The ecological shading treatment significantly affected the community diversity indexes of soil microbial microorganisms (including *Chao1* index, Shannon–Wiener index (*H*), Pielou evenness index (*E*) and Simpson dominance index (*C*)) in the tea plantation (*p* < 0.01 or 0.001; Table 2). When compared with the respective control no-shading treatment (SSCK or EWCK), the ecological shading treatment of WS significantly decreased the *Chao1* index, Shannon–Wiener index (*H*), Pielou evenness index (*E*) and Simpson dominance index (*C*) of the soil microbial microorganisms, while the ecological shading treatments of SS and ES did not significantly affect the community diversity indexes of the soil microbial microorganisms in the tea plantation (Table 2). Moreover, the values of the Shannon–Wiener index (*H*), Pielou evenness index (*E*) and Simpson dominance index (*C*) of the soil microbial microorganisms in the control no-shading treatment of EWCK were significantly higher than those in the control no-shading treatment of SSCK (Table 2).

The *Chao1* index of soil microbial microorganisms in the ecological shading treatment of ES was significantly higher than that in the ecological shading treatment of WS; the Shannon–Wiener index (*H*), Pielou evenness index (*E*) and Simpson dominance index (*C*) of the soil microbial microorganisms in the ecological shading treatment of ES were significantly higher than those in the ecological shading treatments of SS and WS, and these three community diversity indexes of soil microbial microorganisms in the ecological shading treatment of SS were significantly higher than those in the ecological shading treatment of WS (Table 2).

## 4. Discussion

### 4.1. Effects of Ecological Shade by Intercropping Maize on Foliar Soluble Nutrients of Tea Plants

The growth of plants depends on photosynthesis, and the synthesis of plants’ nutrients is inhibited under the condition of insufficient light. However, strong light is one of the environmental factors that leads to a decline in photosynthetic efficiency; the degradation of photosynthetic pigments, photo-oxidative damage, and inactivation of the PSII reaction center which will ultimately lead to the decrease in yield [22,23,24,25,26]. Proper shading could increase the net photosynthetic rate of tea trees and increase the accumulation of photosynthetic products [27]. Although shading reduced the photosynthetic rate of tea for some time, it could alleviate the light inhibition of tea under strong light in summer, which is beneficial to the production of tea [28]. Studies have shown that severe shading reduced the synthesis of carbohydrates in tea and affected the quality of tea. This effect was more severe in the early stage of tea budding [29]. However, in our intercropping design, the shading of corn was a weak effect. In this study, the shading had no significant effect on the contents of soluble sugar and protein in tea. It can be inferred that the maize intercropping had little impact on tea yield. When compared with the maize shading treatment, the differences in tea planting direction were more significant, where the concentrations of soluble sugars and FFA of EWCK were significantly higher than those of SSCK. Maize has a shading effect on tea trees. If the conditions allow, we should measure the illumination of the tea tree crown during the whole growth period of maize, so as to explain the experimental results more reasonably. We failed to measure the light intensity of the tea tree crown throughout the experiment, and only conducted the light intensity measurements in a short period (3d, 6:00 am–6:00 p.m.), which was a weakness of the experiment.

### 4.2. Effects of Maize Intercropped Ecological Shading on Foliar Functional Components and Leaf Quality of Tea Plants

Theanine, caffeine, tea polyphenols and catechins are the key components that affect the bitterness, astringency and freshness taste of tea as a beverage [30]. Studies showed that shading augmented the contents of free amino acids i.e., arginine, glutamic acid and theanine, which are the vital quality determinants of tea [28]. The isotope labeling method showed that dark treatment caused the slow metabolism of theanine and an effective accumulation of theanine [31]. Previous studies found that shading reduced the light intensity and enhanced the activities of enzymes related to the theanine synthesis pathway [32]. The shading environment is dominated by scattered light, and the proportion of blue light was elevated in the tea plantation [32]. The dark respiration of mitochondria is enhanced in blue light, and most flavonoid metabolites decreased significantly in the shading treatments, while the contents of chlorophyll, β-carotene, neoxanthin and free amino acids, caffeine, benzoic acid derivatives and phenylpropanoids increased [33,34].

In the chestnut–tea intercropping, it was found that the foliar contents of amino acids and caffeine were significantly increased, which generally played a positive role in the yield and tea quality [4]. Similarly, it was found that the moderate shading degree of about 50% in summer and autumn had the most positive effect on tea quality, and the foliar contents of amino acids and caffeine increased the most, while the foliar content of tea polyphenols and the phenol/ammonia ratio decreased the most [5]. The ecological shading treatment of SS, WS and ES significantly reduced the tea polyphenol content and increased the caffeine content. The phenol/ammonia ratio and the catechin quality index are crucial indicators for evaluating the quality of green tea and are usually used to determine the suitability for the manufacturing of tea. A low phenol-ammonia ratio and high catechin quality index are more suitable for making green tea [5,30]. In this study, the ecological shading treatment significantly increased the foliar content of theanine and the quality index of catechin in leaves, while decreasing the TP/AA ratio, which had a positive significance for improving the quality of summer and autumn tea.

### 4.3. Effects of Maize Intercropped Ecological Shading on Population Dynamics and the Community Diversity of Key Tea Pests in Tea Plantations

A large number of studies have found that the incidence of plant diseases and insect pests in the compound cultivation mode of tea plantation was relatively mild [35]. When compared with pure tea plantations, ecological tea plantations expand the ecological space and change the ecological environment of tea plantations due to the allocation of multiple species [36]. The existing studies show that the reasonable planting of shading trees in tea plantations is conducive to improving species diversity and the natural control ability of tea plantations, and the ratio of the total number of natural enemies to the total number of pests in shaded tea plantations is higher than that of unshaded tea plantations [35,37]. In the loquat–tea intercropping and citrus–tea intercropping tea plantations, the individual populations of *E. onukii* Matsuda were smaller than those in the pure tea plantations, and intercropping could increase the predatory mite *Anystis baccarum* [14]. Our findings were similar to those of previous studies: as shown in Schedule 1, when compared with the control no-shading treatment, the occurrence of *E. onukii* Matsuda and *T. vaporariorum* (Westwood) in the shade treatments (ES and WS) decreased and the insect diversity index increased. It is speculated that the shading effect of maize might have caused the changes of the content of secondary metabolites in tea leaves, and indirectly affected the occurrence of pests. Studies have shown that secondary metabolites affect insects’ food selection, feeding and utilization, and thus affect insects’ growth, behavior, and population biology [38]. Prior studies mentioned that polyphenols, caffeine and catechins are pest deterrents. Polyphenols have a bitter taste, astringency, and form complexes with soluble protein and are difficult to digest, which can help tea trees resist insect feeding and inhibit insect growth [39,40,41]. On the other hand, maize intercropping increased the number of natural enemy insects.

### 4.4. Effects of Maize Intercropped Ecological Shading on Community Structure and Diversity of Soil Microorganisms in Tea Plantations

Soil microbes decompose organic matter, participate in soil nutrient cycling and plant nutrient supply, and the composition of their community structure is a major indicator for assessing soil quality and fertility [42]. Some scholars believe that the determinants affecting the structural diversity of soil microbial communities include soil type, plant type and soil management measures [43,44]. Current studies shows that when compared with the monoculture system, the alpha diversity of soil bacterial and fungal communities, beta diversity and abundance of the bacterial community were increased in the intercropping system [45], and intercropping can regulate the structural proportion of soil microbial communities [46,47]. An appropriate soil microbial community is beneficial to the sustainable production of tea [48].

The results of this experiment showed that the Chao1 index, Shannon–Wiener index, Pielou evenness index and Simpson dominance index of soil microbial community were decreased by the western shading of maize, i.e., the richness, diversity, evenness and dominance of the soil microbial community were decreased. *Proteobacteria*, *Acidobacteria*, and *Actinobacteria* were the relatively large microflora in farmland soil, and intercropping had an effect on their proportion in the community. In this experiment, the ecological shading of maize affected the composition and distribution of the soil microbial community.

It must be mentioned that the shade was the most intuitive feeling for people when intercropping with corn in tea plantation. However, the causes for the changes of tea detection indexes and quality were closely related to the changes in the soil microbial structure and field temperature, the occurrence of plant diseases and insect pests, and the changes of tea garden biodiversity after the shade formation. The changes in tea quality were the combined result of shading, pest reduction and changes in soil structure. This is an important point in pastoral landscape design: it is not a separate existence but a mutual influence. Agroforestry combines the use of trees with annual crops or fodder plants on the same piece of land that build up synergies, which leads to a higher resilience and allows crops to maintain long-term productivity [49].

## 5. Conclusions

The effects of the ecological shading formed by intercropping maize on tea foliar soluble nutrients, foliar functional components, quality indicators, insect occurrence in tea plantation, and soil microbial diversity were summarized from different planting directions (Appendix A Table A2). The results showed that maize shading in a tea plantation would have little influence on the soluble nutrients of the tea, but would be beneficial to improving the quality of tea. The schematic model also indicates that the eastern and western shading inhibited the occurrence of major pests in tea gardens, which was conducive to enhancing the biodiversity of tea gardens (Figure 9). In addition, ecological shading changed the micro-ecological composition of the soil, and western shading decreased the diversity of soil microorganisms. Thus, intercropping maize in tea plantations is an effective production mode, which affects the field environment from pest occurrence and soil properties, and is advantageous to the tea quality promotion. Further, maize forming the eastern shade is a better choice.

## Figures and Tables

**Figure 1 plants-11-01883-f001:**
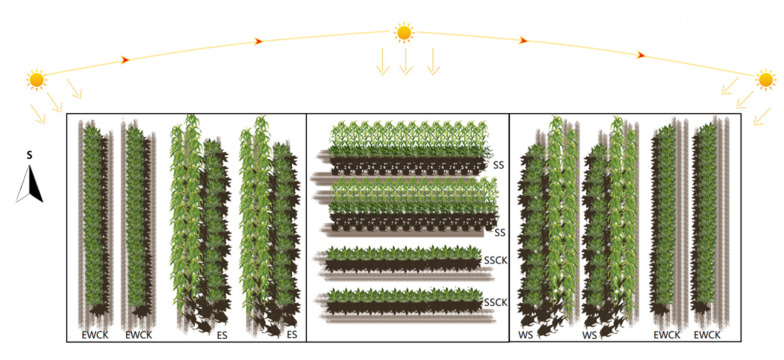
Field layout model of the ecological shading treatments by intercropping maize plants in tea plantation (Note: ES—east shading; WS—west shading; EWCK—control of the WS and ES shading treatments; SS—south shading; SSCK—control of the SS shading treatment).

**Figure 2 plants-11-01883-f002:**
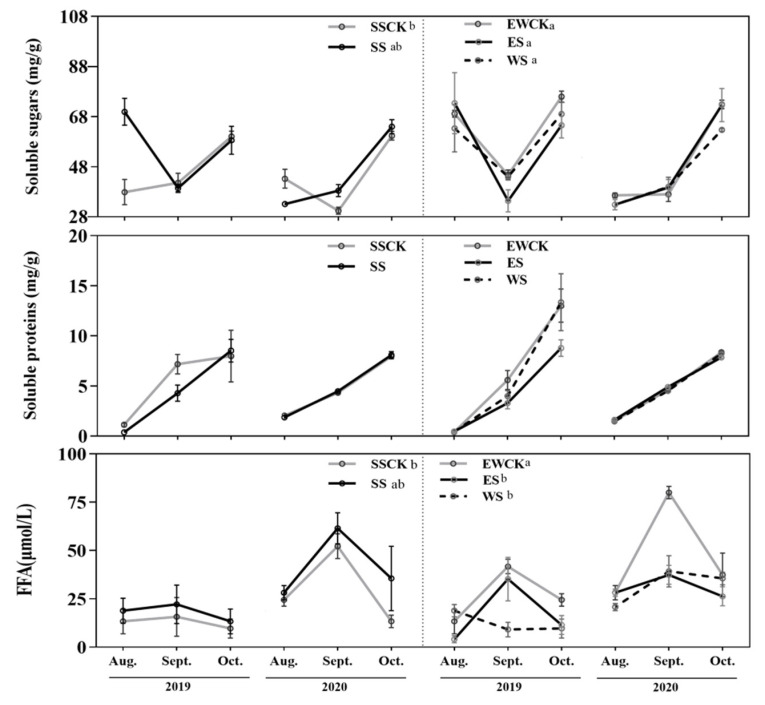
Foliar soluble nutrients of tea plants under intercropped maize ecological shading (**Note:** SSCK, SS, EWCK, ES, WS with at least one identical letter are not significant different from each other by the *LSD* test at *p* > 0.05. The same as in the following figures).

**Figure 3 plants-11-01883-f003:**
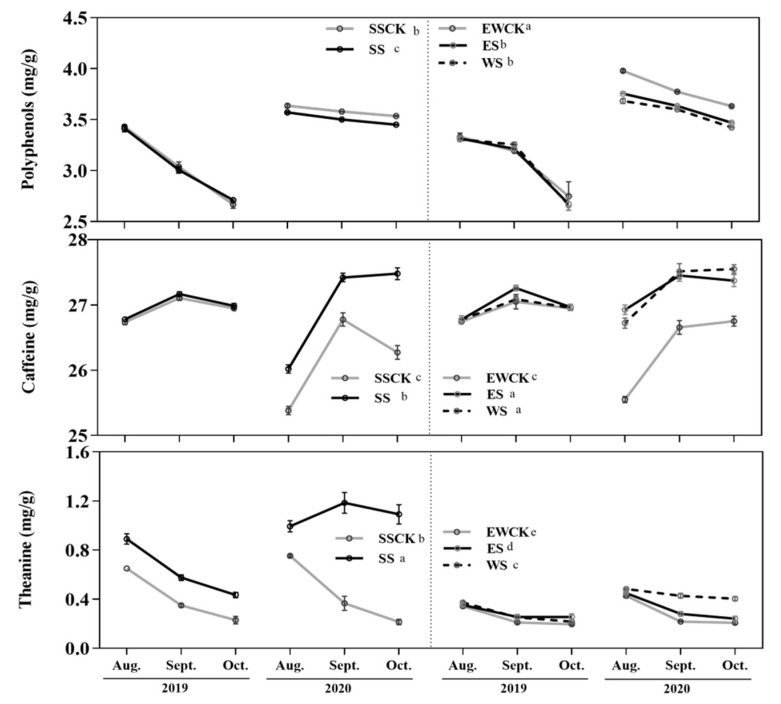
Foliar contents of the functional components of tea plants under intercropped maize ecological shading in a tea plantation.

**Figure 4 plants-11-01883-f004:**
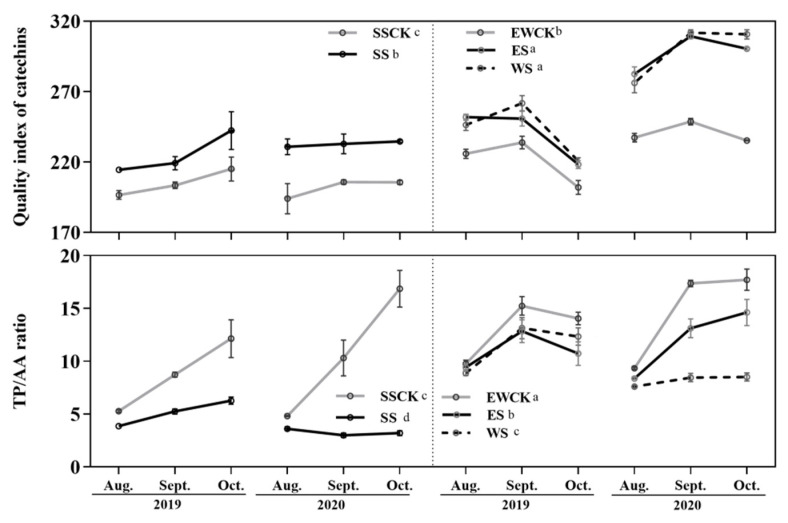
Catechin quality index and phenol/ammonia ratio in the leaves of tea plants under intercropped maize ecological shading in tea plantation.

**Figure 5 plants-11-01883-f005:**
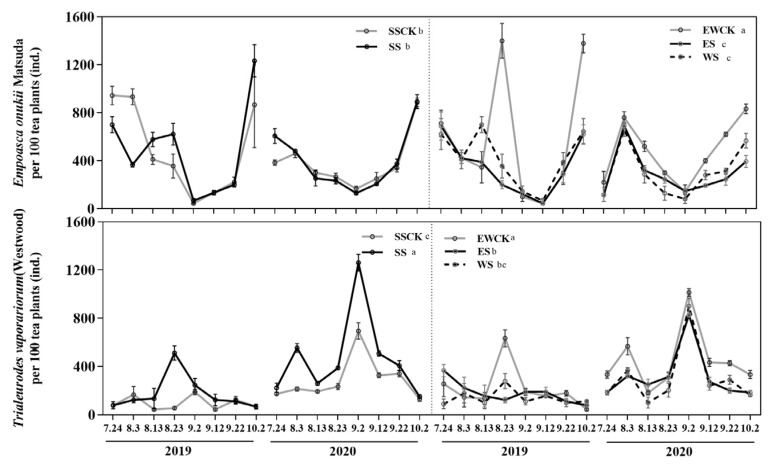
Population dynamics of two key species of insect pests, *Empoasca onukii* and *Trialeurodes vaporariorum* under the intercropped maize ecological shading in tea plantation.

**Figure 6 plants-11-01883-f006:**
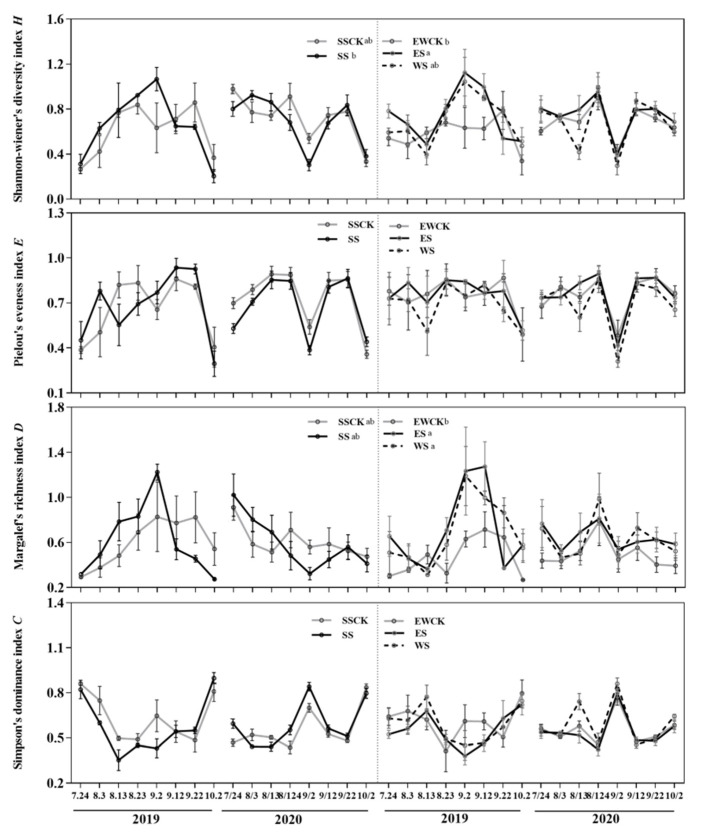
Community diversity indexes of the collected insects under the intercropped maize ecological shading in tea plantation.

**Figure 7 plants-11-01883-f007:**
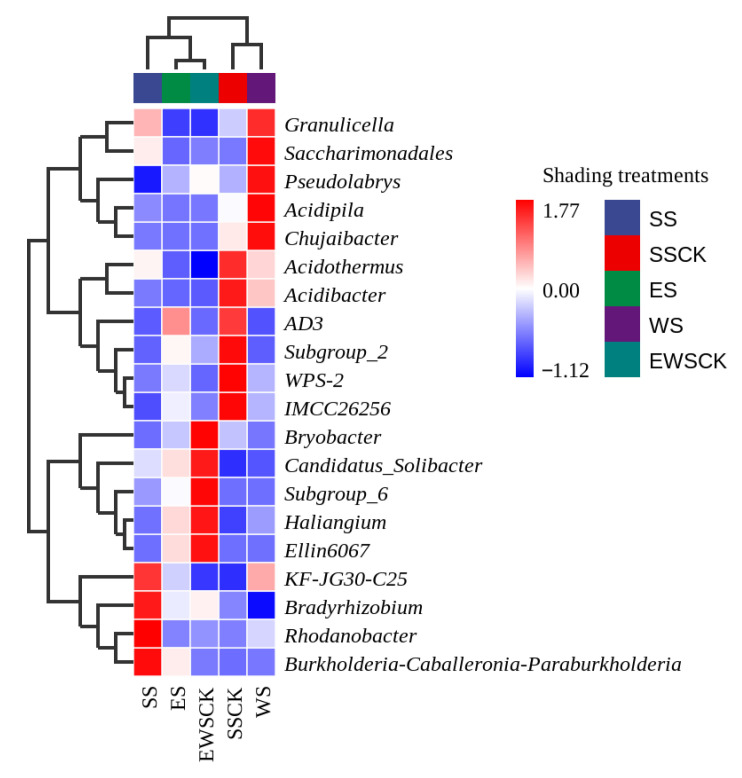
Heat map of the soil microbial composition at genus level under the intercropped maize ecological shading in tea plantation.

**Figure 8 plants-11-01883-f008:**
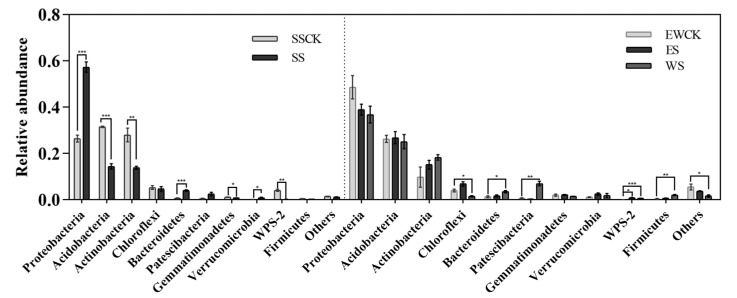
Relative abundance of soil microorganisms at the phylum level under the intercropped maize ecological shading in tea plantation (Note: *, **, and *** show significantly different by the *t* test at *p* < 0.05, *p* < 0.01 or *p* < 0.001).

**Figure 9 plants-11-01883-f009:**
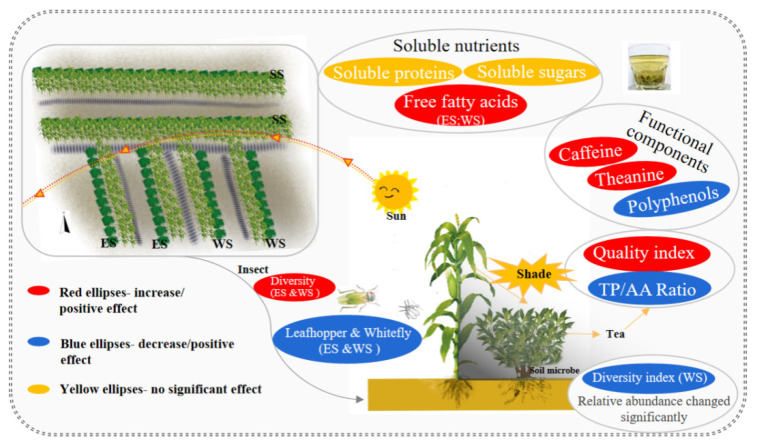
Effects of ecological shading by maize intercropping on foliar soluble nutrients and functional components of tea plants, and population abundances and community diversity of insect pests and soil microbes in tea plantation.

**Table 1 plants-11-01883-t001:** Two-way repeated-measures ANOVAs of ecological shading treatments (including SS, SSCK, ES, WS, EWCK) (S), sampling year (Y) and their interaction (with sampling time as repeated measures) on the foliar contents of soluble nutrients, the functional components and leaf quality indexes of tea plants, and the population dynamics of two key insect species of *Empoasca onukii* and *Trialeurodes vaporariorum*, as well as the community indexes of the collected insects in the tea plantation (values were *F*/*p*) (* *p* < 0.05; ** *p* < 0.01; *** *p* < 0.001).

Measured Indexes		Ecological Shading Treatments (S)	Sampling Years (Y)	S × Y
Foliar soluble nutrients	Soluble sugars (mg/g)	3.83/0.02 *	32.56/<0.001 ***	1.66/0.20
	Soluble proteins (mg/g)	1.73/0.18	2.09/0.16	1.75/0.18
	Free fatty acids (μmol/L)	5.39/0.004 **	53.80/<0.001 ***	0.45/0.77
Foliar functional components	Polyphenols (mg/g)	24.20/<0.001 ***	2331.38/<0.001 ***	13.29/<0.001 ***
	Caffeine (mg/g)	118.21/<0.001 ***	51.33/<0.001 ***	97.48/<0.001 ***
	Theanine (μg/g)	379.93/<0.001 ***	168.34/<0.001 ***	54.42<0.001 ***
Leaf quality	Catechin quality index	151.57/<0.001 ***	171.59/<0.001 ***	35.33/<0.001 ***
	Phenol/ammonia ratio	148.85/<0.001 ***	0.06/0.80	16.06/<0.001 ***
Population dynamics	*Empoasca onukii*	30.61/<0.001 ***	52.62/<0.001 ***	0.73/0.58
	*Trialeurodes vaporariorum*	37.00/<0.001 ***	514.35/<0.001 ***	9.04/<0.001 ***
Community diversity of insects	Shannon-Wiener index (*H*)	2.76/0.04 *	5.15/0.03 *	1.44/0.25
	Pielou evenness index (*E*)	2.64/0.05	1.08/0.31	0.46/0.77
	Margalef richness index (*D*)	2.86/0.04 *	0.25/0.62	0.19/0.94
	Simpson dominance index (*C*)	2.51/0.06	6.34/0.01 *	2.57/0.06

**Table 2 plants-11-01883-t002:** Community diversity indexes of soil microbial microorganisms under the intercropped maize ecological shading in a tea plantation.

Diversity Indices	SSCK	SS	EWCK	ES	WS	F/P
*Chao1* index	6539 ± 61 ab	6556 ± 150 ab	7900 ± 805 a	7784 ± 242 a	5078 ± 222 b	8.42/0.003 **
Shannon-Wiener index (*H*)	10.38 ± 0.07 b	10.23 ± 0.19 b	11.18 ± 0.22 a	11.45 ± 0.03 a	9.31 ± 0.10 c	52.78/<0.001 ***
Pielou evenness index (*E*)	0.840 ± 0.004 b	0.829 ± 0.010 b	0.887 ± 0.104 a	0.898 ± 0.001 a	0.873 ± 0.005 c	43.26/<0.001 ***
Simpson dominance index (*C*)	0.995 ± 0.002 b	0.995 ± 0.001 b	0.999 ± 0.001 a	0.999 ± 0.001 a	0.991 ± 0.001 c	27.43/<0.001 ***

**Note:** ** and *** indicate *p* < 0.01 and *p* < 0.001 by one-way ANOVAs to analyze the effects of ecological shading treatment on the diversity indices of soil microbial communities in the tea plantation, respectively. Means with at least one identical letter are not significant different from each other by the *LSD* test at *p* > 0.05.

## Data Availability

Not applicable.

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
