# Peer review of "Impacts of Intercropped Maize Ecological Shading on Tea Foliar and Functional Components, Insect Pest Diversity and Soil Microbes"

_plants, 2022, doi:10.3390/plants11141883_

Round 1

Reviewer 1 Report

This is an interesting work.

But information of the shading provided by maize-intercropping is lacking. How did the light intensity at the shaded canopy change compared to those of CK? How did the shading effect change with the growth of maize plants? Were the shading effect on different position of the tea canopy change spatially and relate to their impact on quality components? So please provide such information.  

It appears that there were differences between the different directions. Actually three factors were involved in the experimental design: shading, direction, and year. So variance caused by direction has been allocated to and therefore over-estimate the effect of shading by the two-way repeated ANOVA. Heatmap of soil microbial composition (Fig 7)  (and also other results) suggested large difference between EWSCK and SSCK. So it is clear that these are two experiments (at two sites with different layout) and incorrectly combine and analyze them in one ANOVA. I would like to suggest separate dataset into two parallel experiments and re-analyze data. Furthermore, the data included in Table 1 are unclear: F-value and p-level or absolute value with units as shown for indexes?   

It is mentioned that all treatments were replicated for five times. Please provide the area (or dimension) of each plot. How were these plots arranged in the field?

Reviewer 2 Report

The article was shown the effect of ecological shading on tea cultivation through many measurements. However, the relationship between the results of each examination was not clear. The results of this study might be excess a coverage of one article. Therefore, I suggest the article divide into two pieces, the productivity and the ecological aspects of maize shading tea plantation. The other points are followed below.

Introduction

L1,   Camellia sinensis ->  Camellia Sinensis

L42, Why did you choose maize for the ecological shading? Was corn harvested?

Material and Methods

L77, row spacing of 0.10m

The spacing was too narrow.

L86,  .... five times. -> ..... five times (Fig. 1).

L55,  The species of collected insects were also identified and classified.

 Could you show the methods of insect collecting?

Results

L208 and so on   (P = 0.02 and P < 0.001)  -> exclude  

The scale of statistical significance is not important for understanding the text but interferes with reading. 

Table 1

What were shown values in the table? F values? The average of each treatment and the results of multiple comparisons might be more critical.

Figure 2 and so on

Could you show the results of multiple comparisons of each measurement time? What did you think about the seasonal effect? 

Please exclude the lines between Oct. 2019 and Aug. 2020.

Round 2

Reviewer 1 Report

I don't agree  with authors' answer to the Point 1. The main  conclusion is related to the shading effect provided by intercroping with maize whereas there were no data about the light intensity of tea canopy with or without maize. It is very easy to measure the light intensity by a photometer which also is not expensive.  Measurement at different hours on typical days of developmental stages of young shoots is not a difficult thing to do. Lack of light intensity definately  weakens the support to the claim of shading effect.  This is a flaw of experimental design, which makes the conclusion less convincing. 

Author Response

Here, MANY THSNKS for your comments on the measuring light intensity of tea canopy from different shading treatments (ES, WS and SS vs. CK). As carrying out this two-year experiment (2019-2020), we just made the setup of shading treatment, and didn’t measure the light intensity of tea canopy from different shading treatments (This may be one“Black Box” for light intensity). At that time, we thought about using one handheld spectrometer (Model:  ASD FieldSpec® HandHeld 2™) rent from other lab to measure the light intensity, but we indeed didn’t be familiar with this specific tool, so finally we didn’t measure the light intensity of tea canopy from different shading treatments with the intercropped maize plants growing. Based on your comments, we should further carry out some experiments to study the impacts of intercroping with maize or other higher crops or trees on the light intensity of tea canopy in the future study. THANKS AGAIN for your suggestion for us to measure light intensity at different hours on typical days of developmental stages of young shoots of tea plants in tea plantation.

Reviewer 2 Report

Thanks for your comments and revisions. I accept your comments on my review. I have not added any comments since your response was appropriate. I hope this article will be posted soon on Plants.

Author Response

Thank you for your opinions and comments. Wish you success in your work and life. Thank you very much.

Round 3

Reviewer 1 Report

The authors provided mean light intensity measured  on three days without detailed information of measurement. Did the three days cover the whole  period from August 15 to October 15 when samples were taken? Why was light intensity measured only at a few hours from 6:00-10:30 and from 2: 00 am (pm?) to 6: 00 pm ?  The effect observed from the maize intercropping could be caused by other effect than shading.  So change of light intensity is an improtant evidence to support the conclusion (shading effect). Unfortunately such information was incomplete.   In my opinion, it appears that the effect might have been caused combinely by a few  factors including shading, decrease of pests and change of soil properties.  I suggest that authors revise the title, abstract (conclusion) and the maintext accordingly.

Author Response

Point 1: The authors provided mean light intensity measured on three days without detailed information of measurement. Did the three days cover the whole period from August 15 to October 15 when samples were taken? Why was light intensity measured only at a few hours from 6:00-10:30 and from 2: 00 am (pm?) to 6: 00 pm ? The effect observed from the maize intercropping could be caused by other effect than shading. So change of light intensity is an improtant evidence to support the conclusion (shading effect). Unfortunately such information was incomplete.   

Response 1: In the method section, i added the details of the light intensity data measurement method, hoping that it is appropriate. Since it has not measured the light intensity data during the whole experimental period, our supplementary data was an average light intensity at the early stage when the field corn shaded the tea leaves. The measurement of the separation of different time periods was actually the observation results in different shading formation time periods in the field. The local sunrise time and the sunset time during the maize shading stage were about 5: 00 am and 7: 00 pm respectively. We removed the 1-hour measurements of the sun' s rising and near-falling phases because the sun' s light intensity during this time period was small. The solar light intensity was measured every 30 minutes according to the actual shading conditions in the field, as a supplement to the light intensity data between different shading treatments and the control. It may not be perfect, but it is representative. Hope to get your approval.

Point 2: In my opinion, it appears that the effect might have been caused combinely by a few  factors including shading, decrease of pests and change of soil properties.  I suggest that authors revise the title, abstract (conclusion) and the maintext accordingly.

Response 2: I quite agree with your point of view. In field experiments, any input of matter or energy can change the whole environment. In our experimental design, the artificial variable was not only the shade brought by corn, but also the impact of corn on the environment. I have already made a brief discussion in the discussion section to illustrate this point.